# Meta-Analysis of Heat-Stressed Transcriptomes Using the Public Gene Expression Database from Human and Mouse Samples

**DOI:** 10.3390/ijms241713444

**Published:** 2023-08-30

**Authors:** Sora Yonezawa, Hidemasa Bono

**Affiliations:** 1Laboratory of Genome Informatics, Graduate School of Integrated Sciences for Life, Hiroshima University, 3-10-23 Kagamiyama, Higashi-Hiroshima City 739-0046, Japan; m224825@hiroshima-u.ac.jp; 2Laboratory of BioDX, Genome Editing Innovation Center, Hiroshima University, 3-10-23 Kagamiyama, Higashi-Hiroshima City 739-0046, Japan

**Keywords:** heat stress, meta-analysis, public databases, gene expression, RNA-seq, ChIP-seq analysis

## Abstract

Climate change has significantly increased the frequency of our exposure to heat, adversely affecting human health and industries. Heat stress is an environmental stress defined as the exposure of organisms and cells to abnormally high temperatures. To comprehensively explain the mechanisms underlying an organism’s response to heat stress, it is essential to investigate and analyze genes that have been under-represented or less well-known in previous studies. In this study, we analyzed heat stress-responsive genes using a meta-analysis of numerous gene expression datasets from the public database. We obtained 322 human and 242 mouse pairs as the heat exposure and control data. The meta-analysis of these data identified 76 upregulated and 37 downregulated genes common to both humans and mice. We performed enrichment, protein–protein interaction network, and transcription factor target gene analyses for these genes. Furthermore, we conducted an integrated analysis of these genes using publicly available chromatin immunoprecipitation sequencing (ChIP-seq) data for HSF1, HSF2, and PPARGC1A (PGC-1α) as well as gene2pubmed data from the existing literature. The results identified previously overlooked genes, such as *ABHD3*, *ZFAND2A*, and *USPL1*, as commonly upregulated genes. Further functional analysis of these genes can contribute to coping with climate change and potentially lead to technological advancements.

## 1. Introduction

Climate change is expected to result in higher average temperatures and an increased intensity and frequency of heat waves, thereby increasing the adverse effects of high temperatures. For example, it has been reported that the average temperature on land where humans reside was projected to be 0–6 °C higher in the summer of 2021 than that in the years 1986–2005. Moreover, an increase in the number of heat-wave days experienced by children under the age of 1 year and adults over the age of 65 years, which are the age groups vulnerable to high temperatures, has been observed. These phenomena are anticipated to cause heat-related diseases (e.g., heat stroke) and are considered as indicators of the adverse effects of climate change on human health [1,2,3]. Visual climate change data are available at https://www.lancetcountdown.org/data-platform/ (accessed on 1 August 2023).

High-temperature environments impose stress on organisms and cause various cellular changes. Heat stress (HS, also known as heat shock or hyperthermia) is an environmental stress that occurs when organisms and cells are exposed to abnormally high temperatures, which are thought to increase with climate change. HS has diverse effects on the elements that form cells [4]. At the cellular level, it causes the uncoupling of oxidative phosphorylation in the mitochondria, which are cellular organelles, and promotes the generation of reactive oxygen species (ROS) [5,6]. Additionally, it causes physical changes in the biomembrane, such as changes in membrane fluidity [7]. HS exerts various effects at the molecular level; for example, it can inhibit DNA repair pathways and cause DNA damage [8,9,10]. HS can also denature intracellular proteins and lead to their aggregation [11].

A programmed gene expression response called the heat shock response (HSR) is a cellular response to HS. The HSR is primarily regulated by a group of transcription factors called heat shock factors (HSFs); of them, HSF1 plays a central role in the response. When triggered by HS, HSF1 forms a trimer and binds to heat shock elements (HSEs) in DNA, which induces the expression of genes encoding heat shock proteins (HSPs). The main HSP members are important proteins that function as molecular chaperones and maintain protein homeostasis (proteostasis) by interacting with other proteins, stabilizing them, and helping them to acquire functional conformations. These members are mainly classified as HSP40, HSP60, HSP70, HSP90, HSP100, and sHSPs (small HSPs), and they allow cells to maintain proteostasis and survive [12,13,14].

To date, HS-related studies have primarily focused on proteins that function as molecular chaperones, accumulating knowledge regarding their role in maintaining cell proteostasis and protecting cells under stress conditions, including heat stress. However, heat affects numerous cellular components [4]; moreover, proteins induced by heat stress have been proposed to be classified into functional classes such as nucleic acids and metabolism [12]. Despite this classification, limited studies have been available on the genes encoding these proteins induced by HS. Therefore, a comprehensive analysis, including previously uncharacterized or poorly known HS-related genes, is essential to deeply understand the heat stress response.

One approach to solving this problem is data-driven research using meta-analysis. Meta-analysis is a research method that integrates the results of multiple studies to obtain new insights. Specifically, a meta-analysis of gene expression data has been made possible by the increasing amount of data accumulated in public databases and by developing comprehensive methods for analyzing data from multiple studies [15]. This approach has led to the meta-analysis of gene expression data focused on various phenomena and species.

For example, meta-analyses incorporating a medical perspective have focused on hypoxia and oxidative stress. These studies collected, integrated, and analyzed human transcriptome data, abundant in public databases, to identify novel genes and pathways involved in responses to these specific stressors [16,17]. The importance of meta-analysis has also been pointed out in basic biology [18]. In addition to contributing to medical insight research, studies focusing on hypoxia in rice and Arabidopsis [19], as well as density-dependent polyphenism in insects [20], have demonstrated the versatility of meta-analysis, expanding its application across a diverse range of organism species and phenomena. Furthermore, attempts are being made to combine gene expression data with data from ChIP-seq and previous information from the existing literature to gain deeper insights [15,16].

Given this background, this study aimed to analyze genes involved in heat stress (HS) by collecting HS-related gene expression data from public databases on humans and mice and performing a meta-analysis. Unlike previous hypothesis-driven studies, this approach may yield novel insights into the mechanisms underlying HS. The dataset collected in this study and the selected gene set will contribute to a more comprehensive understanding of organisms’ responses to HS.

## 2. Results

### 2.1. Overview of Analysis Scheme

The basic scheme of the analysis is shown in Figure 1. In Section 2.2, “Characteristics of HS-related High-Throughput Sequencing Data”, we describe the collection of gene expression data from public databases. Section 2.3, “Verification of Upregulated and Downregulated Genes in Each Organism”, details the calculation of HN-ratios and the HN-score, along with the extraction and evaluation of gene groups. In Section 2.4, “Identification of Common Upregulated or Downregulated Genes in Human and Mouse”, common genes are identified and subjected to various analyses, including enrichment analysis. Section 2.5, “Integration of HN-score, Transcription Factor-binding Information, and Literature Information”, integrates and analyzes ChIP-seq data, as well as gene2pubmed data as information from the existing literature.

### 2.2. Characteristics of HS-Related High-Throughput Sequencing Data

In this study, we obtained 66 Sequence Read Archive (SRA) IDs and a total of 564 pairs of gene expression data. The gene expression data consisted of 322 pairs of human and 242 pairs of mouse data, which compared HS and non-treatment (control) conditions. We searched for RNA sequencing (RNA-seq) data but encountered reports using different sequencing methods. Therefore, we collected gene expression data for various sequencing methods, such as RNA-seq and Ribo-seq (also known as ribosome profiling) to obtain a broader dataset. A summary of the sample types, temperature conditions, and treatment times for the collected data is shown in Figure 2. In the human samples, primarily cultured cells were collected, while in the mouse samples, both cultured cells and tissues were collected. The samples with the most data included cultured human cells (MRC5-VA: 55 pairs), mouse cells (Mouse embryonic fibroblast (MEF) cells: 52 pairs), and mouse tissues (kidneys: 23 pairs) (Figure 2A). In the human samples, 172 pairs (30.5%) were derived from cancer cells. The most common conditions were a temperature of 42 °C for the human and mouse samples, and a treatment time of 60 min for the human samples and 30 min for the mouse samples (Figure 2B,C). The conditions included in the “other” category comprised samples with discontinuous treatment times, such as a sample in which the temperature was changed midway from 42 °C to 48 °C, or samples exposed to high-temperature conditions repeatedly for 3 h per day for over 1 week (Figure 2B,C). Metadata, including detailed information on temperature conditions, sample types, sequencing methods, etc., are available in the figshare file (Appendix A) [21] and provide a background on the collected data.

### 2.3. Verification of Upregulated and Downregulated Genes in Each Organism

After quantifying the expression using the analysis pipeline ikra [22], expression ratios (HN-ratios) were calculated for the data collected in pairs of HS and non-treatment conditions. The HN-ratios calculated for all genes were then classified as “upregulated”, “downregulated”, or “unchanged” if neither threshold was met, according to a defined threshold (5-fold or 1/5-fold in this study). This classification facilitates the interpretation of complicated gene expression data [15]. Next, the HN-score was calculated for all genes as an index of analysis to detect genes with variable expression. The HN-score is the number of study data classified as “upregulated” minus the number of study data classified as “downregulated”. The HN-scores for all human and mouse genes are visualized in Figure 3A,D scatter plots, respectively. Using the gene (HSPA6) with the highest HN-score in Figure 3A as an example, the data counted as “upregulated” 247 times, “downregulated” 2 times, and “unchanged” 73 times; the HN-score is calculated to be 245 (247-2). HN-ratios and HN-scores for all human and mouse genes, respectively, are available in the figshare file (Appendix A) [21].

Gene sets were extracted based on HN-score, and a gene set enrichment analysis was performed using Metascape [23]. The top 500 genes with high HN-scores and the bottom 500 genes with low HN-scores were extracted from the human and mouse samples, respectively. The top 500 genes were designated as the “upregulated gene group” (red dots in Figure 3A,D), while the bottom 500 genes were referred to as the “downregulated gene group” (blue dots in Figure 3A,D). In the upregulated gene group, “protein folding” (Gene Ontology (GO):0042026) was the most significantly enriched GO term for humans and mice (Figure 3B,E). In contrast, in the downregulated gene group, “chromosome organization” (GO:0051276) was the most enriched GO term in human samples, while “mitotic sister chromatid segregation” (GO:0000070) was the most enriched GO term in mouse samples (Figure 3C,F).

### 2.4. Identification of Common Upregulated or Downregulated Genes in Human and Mouse

From the obtained upregulated and downregulated human and mouse gene sets, we extracted sets of genes common to both human and mouse samples to analyze the genes with a typical response to HS. We accessed Mouse Genome Informatics (MGI) [24], a database that compiled a variety of information on mice, and used the Batch Query function to convert the gene symbols of the human upregulated and downregulated gene sets into the corresponding mouse ortholog gene symbols denoted by “Current Gene Symbol” (e.g., *HSPA1A* → *Hspa1a*). Consequently, 410 upregulated genes and 406 downregulated genes from the human set were converted into mouse orthologs. Of them, 76 genes were commonly upregulated, and 37 were commonly downregulated (Figure 4A,B). A scatter plot was created to evaluate the distribution of the HN-scores for the common genes using the HN-scores of the corresponding orthologous human and mouse genes (Figure 4C). In the scatter plot, the 76 commonly upregulated genes are shown in red, and the 37 commonly downregulated genes are shown in blue. As shown in Figure 4C, a notable pattern was observed for *HSPA1A* and *HSPA1B*, which encode the HSP70 family of proteins. Various stressors, including HS, induce the expression of these genes. Using ShinyGO (ver. 0.77) [25], the common upregulated and downregulated genes were plotted against the human genome, revealing their distribution (figshare file Appendix A) [21]. Among the 76 common upregulated genes, 10 genes, including *HSPA1A* and *HSPA1B*, were annotated as “response to heat” (GO:0009408). The list of upregulated and downregulated genes with symbols and the HN-score data for humans and mice are available in the figshare file (Appendix A) [21]. In addition, HN-scores were calculated for each sample, sequencing method, and experimental condition (temperature and exposure time) and represented in a stacked bar graph (figshare file Appendix A) [21].

We performed a gene set enrichment analysis with human gene symbols using Metascape [23], as previously conducted in the “Verification of upregulated and downregulated genes in each organism” section, to characterize the 76 common upregulated genes and 37 common downregulated genes (Figure 5A,B). Additionally, the search results using mouse gene symbols are shown in the figshare file (Appendix A) [21]. Among the common upregulated genes, “response to topologically incorrect protein” (GO:0035966) was the most enriched (Figure 5A). Within this term, the following genes were found: (1) heat shock 70 kDa proteins (HSPA): *HSPA1A*, *HSPA1B*, *HSPA1L*, *HSPA4L*, *HSPA8*, *HSPH1*; (2) DNAJ (HSP40) heat shock proteins: *DNAJA1*, *DNAJA4*, *DNAJB1*, *DNAJB4*; (3) small heat shock proteins (HSPB): *HSPB1*, *HSPB8*; (4) chaperonins: *HSPD1*, *HSPE1*; (5) heat shock 90 kDa proteins (HSP90): *HSP90AA1*; (6) BAG cochaperones (BAG): *BAG3*; and (7) serpin peptide inhibitors (SERPIN): *SERPINH1* (also known as hsp47). Thus, the term covers genes encoding various classes of molecular chaperones and cofactors, and these genes, which play a role in maintaining proteostasis, are commonly expressed in both humans and mice. Alternatively, the term “negative regulation of chromosome organization” (GO:2001251) was found to be the most enriched among the common downregulated genes (Figure 5B). This term includes genes that play essential roles in the regulation of mitosis, such as *PLK1* and *HASPIN*.

Protein–protein interaction networks were analyzed for more detailed information on the commonly upregulated and downregulated genes annotated in the enriched GO terms (Figure 5C,D). For common upregulated genes, a cluster of genes encoding molecular chaperones was visualized using the Molecular Complex Detection (MCODE) algorithm (Figure 5C; shown in red). Groups of genes classified as immediate early genes (IEGs), including *FOS*, *FOSB*, and *JUN*, which encode transcription factor activator protein 1 (AP-1), were also clustered (Figure 5C; shown in blue). IEGs are a group of genes that are rapidly expressed in response to cellular stimuli. The *FOS* and *JUN* genes extracted in this study have been well-characterized in previous studies. *EGR1*, *EGR2*, *FOS*, *FOSB*, and the *ARC* of IEGs in this blue cluster were annotated in the “NGF-stimulated transcription” pathway (R-HSA-9031628). Additionally, actin alpha 1 skeletal muscle (*ACTA1*), which belongs to the actin family, was also found in the protein–protein interaction network, although it was not included in the cluster. In contrast, no closely linked networks were formed for the downregulated genes compared to those observed for the upregulated genes (Figure 5D).

We conducted an enrichment analysis to obtain an overview of the gene expression regulatory information for common upregulated genes and common downregulated genes (Figure 5E,F). Gene sets were obtained from the Molecular Signatures Database (MSigDB) [26]. Among the commonly upregulated genes, PPARGC1A_TARGET_GENES (systematic name M30124) was the most enriched (Figure 5E). *PPARGC1A* encodes peroxisome proliferator-activated receptor gamma coactivator 1α (PGC1-α), a critical regulator of mitochondrial biogenesis. Terms related to HSF1 and HSF2, the central regulators of HS, such as HSF2_TARGET_GENES (systematic name: M30020), TTCNRGNNNNTTC_HSF_Q6 (systematic name: M16482), and RGAANNTTC_HSF1_01 (systematic name: M8746), were also included. Genes common to these four sets, such as *HSPA1A*, which encodes a molecular chaperone, were also included. The common genes are available on figshare (Appendix A) [21]. Based on these results, we conducted a detailed analysis of HSF1, HSF2, and PPARGC1A. A gene set related to serum response factor (SRF), SRF_C (systematic name: M12443), was also found in the enrichment analysis, although it was not as enriched (Figure 5E). This gene set contained IEGs (*FOS*, *FOSB*, *EGR1*, and *EGR2*) included in the blue cluster shown in Figure 5C, and *ACTA1*, which was not included in the blue cluster. On the other hand, in the common downregulated genes, the enrichment analysis results were inconclusive (Figure 5F). This observation, along with the strong enrichment of target genes by transcription factors such as HSF1, HSF2, and PPARGC1A in common upregulated genes (Figure 4E), led to our decision to focus further analysis on common upregulated genes.

### 2.5. Integration of HN-Score, Transcription Factor-Binding Information, and Literature Information

In the previous section, we further explored the enrichment analysis of transcription factor target genes. This analysis revealed that the target genes of HSF1, HSF2, and PPARGC1A were strongly enriched in the common upregulated genes compared to the common downregulated genes (Figure 5E,F). To further analyze the binding information of these three transcription factors, we integrated ChIP-seq data from the ChIP-Atlas database [27] processed using the MACS2 program, with the HN-score results obtained in this study. In ChIP-Atlas, the ChIP-seq data from multiple studies are integrated and analyzed, and MACS2 scores for each gene can be obtained by querying the transcription factors and the range around the transcription start sites [27]. Scatter plots illustrating the ChIP-seq and HN-scores were used for visualization (Figure 6). Additionally, we used gene2pubmed data to visualize the number of reports registered in PubMed for each of the 76 common upregulated genes. The source data can be obtained from the figshare file (Appendix A) [21]. As shown in Figure 6A, in addition to HSP genes such as *HSPA1A*, which are known to be targets of HSF1, abhydrolase domain-containing 3 (*ABHD3*) was identified as a target candidate among genes with fewer research reports. In contrast, genes with low ChIP-seq scores were observed. *EGR1* and *ACTA1*, the target genes of SRF (Figure 5C,E), showed such characteristics, suggesting that they belong to pathways unrelated to HSF1. Common genes with high ChIP-seq scores were found not only for HSF1 (Figure 6A), but also for HSF2 (Figure 6B) and PPARGC1A (Figure 6C). In addition to the HSP genes, zinc finger AN1-type containing 2A (*ZFAND2A*) and ubiquitin-specific peptidase-like 1 (*USPL1*), had high ChIP-seq scores (Figure 6).

## 3. Discussion

In this study, we manually curated 322 human and 242 mouse pairs of heat-exposed and untreated sample data from public databases for gene expression data, and performed HS-related gene analysis. The top 500 genes with the highest HN-scores in humans and mice were selected as upregulated genes, and the bottom 500 were selected as downregulated genes for the gene set enrichment analysis. GO terms associated with protein folding were the most enriched in the upregulated gene dataset for both species (Figure 3A,D). Considering that HS disrupts proteostasis [4,12], this gene set seems to reflect the variation in gene expression under HS conditions reported in previous studies. It is important to note that this study did not take comprehensive statistical methods, including gene extraction, so interpretation should be done with caution. However, our approach using the HN-ratio and HN-score suggests the potential to extract HS-related genes, though further validation may be required.

To identify common heat shock response (HSR) mechanisms, we narrowed down our results to the upregulated and downregulated genes common to both humans and mice and identified 76 upregulated genes and 37 downregulated genes (Figure 4A,B). Detailed information on these two gene sets was obtained via protein–protein interaction network analysis, transcription factor target gene enrichment analysis, and gene set enrichment analysis (Figure 5). Several genes encoding molecular chaperones, including *HSPA1A* and *HSPA1B*, were included in the common upregulated genes, and clusters of these genes were created (Figure 5A,C). Results from the meta-analysis have also revealed this response to proteostasis induced by HS. In contrast, genes associated with different processes, such as IEGs and *ACTA1*, were also present in the common upregulated genes (Figure 5C). These genes are also targets of the transcription factor SRF, which is distinct from HSF1, the master regulator of the HSR (Figure 5E). It is already known that SRF is associated with the expression of IEGs (e.g., *FOS*) [28]. In addition, genome-wide analysis has revealed that SRF mainly regulates genes induced early in HS in an HSF1-independent manner, and many of these genes are related to the cytoskeleton [29]. Therefore, genes associated with HSF1-independent pathways were identified. 

Moreover, in addition to the HN-score data for each gene calculated in this study, the ChIP-seq data as transcription factor-binding information and the gene2pubmed data as information from the existing literature were integrated to visualize the characteristics of each gene, and information on the common upregulated human and mouse genes was added (Figure 6). In the ChIP-Atlas database, registered ChIP-seq data were scored using the MACS2 score program, with higher scores suggesting a direct binding of transcription factors [27]. This information can be used to filter genes that are directly or indirectly regulated by the transcription factor of interest, in addition to knowing the potential target genes of the transcription factor [15]. As shown in Figure 6A, genes directly regulated by HSF1, the master regulator of the response to HS, as well as genes thought to be indirectly regulated, were included in the common upregulated genes. In addition to HSF1, we targeted HSF2, a member of the HSF family, and PPARGC1A (PGC1-α), which is associated with various metabolic events because of the enrichment of genes encoding different transcription factors (Figure 5E). HSF2 interacts with HSF1, forming a heterodimer with HSF1 under HS conditions [13,30]. On the contrary, PPARGC1A (PGC1-α) may be associated with the induction of the expression of representative HSP genes in the red cluster in Figure 5C and has been shown to activate *Hspa1a* transcription by interacting with HSF1 in mouse 10T1/2 cells [31]. Thus, the regulation of gene expression in response to HS may be co-ordinated with transcription factors other than HSF1. However, further investigation is needed because it has not been possible to identify all the transcription factors involved in transcriptional regulation in response to heat stress. The results of this study also suggest that this is a common gene with a high ChIP-seq score, as shown in Figure 6. Thus, the integration and use of transcription factor-binding information suggest that it may be a helpful method for a more detailed analysis of the response to HS.

In addition to the information on transcription factor regulation using the ChIP-seq data, gene2pubmed data were used to assess gene attention and visualize the results (Figure 6). We investigated 76 common upregulated genes with high ChIP-seq scores. As shown in Figure 6A, *ABHD3*, with a high ChIP-seq score, was implicated in the degradation of oxidatively truncated PCs (oxPCs) generated by oxidative stress as well as medium-chain phospholipids [32]. It has been suggested that heat alters biomembranes [7]. Heat can also affect mitochondria, which are cellular organelles, and may contribute to the production of ROS [5]. *ABHD3* may contribute to biomembrane homeostasis by removing oxPCs generated by ROS during HS. In contrast, we found that *ZFAND2A* and *USPL1* had high ChIP-seq scores for the three transcription factors (Figure 6A–C). *ZFAND2A* is a gene whose expression is induced by heat, and HSF1 has been suggested to be involved in this process [33]. Notably, there have been few reports of this gene, despite reports of its association with HS. These examples indicate that HS research focuses on the expression and functional analysis of molecular chaperones. However, there is no current evidence suggesting that *USPL1* is associated with HS. It has been suggested that *USPL1* encodes a protein that does not target ubiquitin but functions as a small ubiquitin-related modifier (SUMO) isopeptidase [34]. However, according to the gene2pubmed data, there have been very few reports of this (20 articles published as of April 2023), and its functions still need to be fully understood.

While comprehensive analyses and new candidate HS-responsive genes have been selected, limitations of this study exist. Broadly categorized, there are (1) biases due to data from diverse backgrounds, including sequencing methods, cell types, and experiment conditions, and there is also (2) the interpretation of the results from a biological perspective. Regarding the former issue (1), the transparency of the study was improved by disclosing metadata about the collected data (Appendix A) and by visually representing the contribution of each sample to the HN-score in a stacked bar graph (Appendix A) [21]. However, caution should be exercised in interpreting the results, as there may still be bias due to factors not considered, such as conditions not mentioned in the metadata. In viewpoint (2), several points must be made. For example, this is an area where the complexity of the HSR is not captured. Some mechanisms have not been focused on, such as the perspective of the hierarchical levels of regulation in multicellular organisms [12] and the regulation of expression by transcription factors that have not been identified in this study. Furthermore, careful interpretation must include considerations such as the backgrounds (genetic, physiological, and evolutionary aspects) that humans and mice do not share.

In addition to addressing these research limitations, a detailed functional analysis of genes (such as *USPL1*), whose response mechanisms and functions have not been elucidated in detail, could further expand the scope of heat stress research. Further functional analysis should include the use of genome editing tools such as CRISPR-Cas9, which enable the precise manipulation of target genes and are especially useful for the functional analysis of genes whose functions are unknown for the HS data extracted in this study. Furthermore, by utilizing tools such as Metascape [23], which integrates various information resources (Gene Ontology, KEGG Pathways, etc.) to provide multifaceted insights into the functional significance of the identified gene expression changes, and facilitate more targeted experimental validation, it is expected that this study will serve as a basis for subsequent in-depth functional analysis studies. Therefore, the genes identified and validated in this study may serve as candidates for novel genome-editing target genes.

In this study, a meta-analysis of the human and mouse gene expression data facilitated the extraction of genes not yet known to be associated with HS, in addition to the responses of organisms to heat derived from previous findings. Such findings from data-driven studies can potentially provide novel insights. However, it is unclear whether the genes identified in this study are as widely conserved in species as molecular chaperone genes; therefore, a meta-analysis of gene expression data on a broader range of species should be conducted to identify novel common mechanisms of HS responses. This knowledge could contribute to addressing the increased exposure to and frequency of high temperatures due to climate change. With growing concerns about increased health hazards and impacts on industries such as crop production [1], this research lays the groundwork for developing countermeasures to these problems and may lead to new knowledge and technological advances in future research.

## 4. Materials and Methods

### 4.1. Curation of Gene Expression Data from Public Database

The public database Gene Expression Omnibus (GEO) [35] was used to obtain gene expression data associated with human and mouse HS. GEO is operated by the National Center for Biotechnology Information (NCBI) and archives gene expression data obtained by RNA sequencing (RNA-seq) using Next-Generation Sequencing (NGS) and expression microarrays. To narrow down the gene expression data, we used keywords related to heat exposure such as “heat stress”, “heat shock”, “hyperthermia”, “thermal stress”, “heat-shock treatment”, and “heat stroke”. Additionally, “Expression profiling by high throughput sequencing” was added to the search formula as a study-type condition. Data collection was done manually. Data from heat stress and non-treatment conditions were collected if they could be made as a pair or if the data were from a study examining heat stress, and if deemed appropriate.

### 4.2. Quantification of Gene Expression Data Using the Analysis Pipeline

The gene expression data retrieval, processing, and quantification were performed using ikra (ver. 2.0.1) [22], an automated human and mouse RNA-seq data analysis pipeline that can automatically perform all of the following processes: retrieve FASTQ format files using the fasterq-dump program in the NCBI SRA tool kit (ver. 3.0.0) [36], read quality control, and read trimming using Trim_galore (ver. 0.6.6) [37] as well as transcript quantification using Salmon (ver. 1.9.0) [38]. Pre-processing through Trim_galore, the step prior to expression quantification, strictly filters out low-quality reads and adapter sequences that could affect results, ensuring the reliability and accuracy of the gene expression data curated [37]. This study used a different version of the tool compared to that used in previous studies [16,17]: the SRAtoolkit, Salmon. Different versions of the reference sequence sets that were used as indices in Salmon were also used, including GENCODE Release 37 (GRCh38.p13) for the human data and GENCODE Release 26 (GRCm39) for the mouse data.

### 4.3. Calculation of HN-Ratio

Expression ratios (henceforth referred to as the HN-ratio) were calculated for all genes from the gene expression data paired with HS and non-treatment. The HN-ratio for each gene was calculated using the following Equation (1)
(1)$$HN-ratio=\frac{T_{HS}+1}{T_{non-treatment}+1}$$

*T_HS_* and *T_non-treatment_* refer to scaled Transcripts Per Million (scaled TPM) [39] under HS and non-treatment conditions, respectively, and indicate the quantified expression levels. When calculating the HN-ratio, 1 was added to the expression value to avoid calculation with a zero value [17,40].

### 4.4. Classification of Genes Based on HN-Ratio

Using the calculated HN-ratio, all genes were classified into three groups: upregulated, downregulated, and unchanged. If the HN-ratio was higher than a threshold, the gene was considered “upregulated”. Conversely, if the HN-ratio was less than the reciprocal of a threshold, the gene was considered “downregulated”. Genes not classified in either group were considered “unchanged”. For genes that were upregulated, we tested 2-fold, 5-fold, and 10-fold thresholds and selected the 5-fold threshold; for genes that were downregulated, we tested 1/2-fold, 1/5-fold, and 1/10-fold thresholds and selected the 5-fold threshold for classification.

### 4.5. Calculation of HN-Score

To evaluate the genes whose expression was altered by HS, an index called the heat stress and non-treatment score (HN-score) was calculated for each gene in humans and mice. The HN-score was calculated by subtracting the number of instances of genes classified as downregulated from those of genes classified as upregulated. The HN-ratio and HN-score were calculated using a script from a previous study (https://github.com/no85j/hypoxia_code, accessed on 1 August 2023) [16]. Python scripts were created to visualize the HN-score values for each gene using a scatter plot (https://github.com/yonezawa-sora/HS_code, accessed on 1 August 2023).

### 4.6. Analysis of Selected Gene Sets in Human and Mouse

From each human and mouse dataset, the top and bottom genes were selected based on the HN-score, and a gene set enrichment analysis was performed using the web tool Metascape [23] (accessed November 2022). The Batch Query function of mouse genome informatics [24] (accessed November 2022) was used to extract genes that are commonly conserved between humans and mice. Scatter plots of the HN-scores of human genes corresponding to the mouse “Current gene symbol” and the HN-scores of mouse genes were created using a Python script. The scripts used are available on GitHub (https://github.com/yonezawa-sora/HS_code, accessed on 1 August 2023). Gene set enrichment analysis, protein–protein interaction networks, and transcription factor target gene set enrichment analyses were performed using Metascape [23]. The protein–protein interaction networks were processed using Cytoscape [41]. The genome mapping was performed using ShinyGO [25] (ver. 0.77) (accessed February 2023). A Python script was used to create a stacked bar graph to visualize the contribution of each sample to the HN-score. The script is available on GitHub (https://github.com/yonezawa-sora/HS_code, accessed on 1 August 2023).

### 4.7. Comprehensive Analysis of Common Upregulated Genes in Human and Mouse

The ChIP-Atlas Database [27] (accessed February 2023) was accessed, and the “Target Genes” function was used to retrieve an average model-based analysis of ChIP-seq (MACS2) scores for all genes. HSF1, HSF2, and PPARGC1A (PGC1-α) were used as “antigens”. MACS2 scores were obtained at distances of ±5 kb from the transcription start site. The numbers of the ChIP-seq experiment data (accessions are indicated by SRX ID) used to calculate the average MACS2 score are HSF1:105, HSF2:25, and PPARGC1A:1. For genes for which a MACS2 score was not calculated by ChIP-Atlas, this MACS2 score was considered to be 0. For each gene, the calculated HN-score, MACS2 score, and the number of reports in the existing literature on the human gene in the gene2pubmed data (accessed February 2023) were visualized by creating a scatter plot using a Python script. The script used for the visualization is available on GitHub (https://github.com/yonezawa-sora/HS_code, accessed on 1 August 2023)).

## Figures and Tables

**Figure 1 ijms-24-13444-f001:**
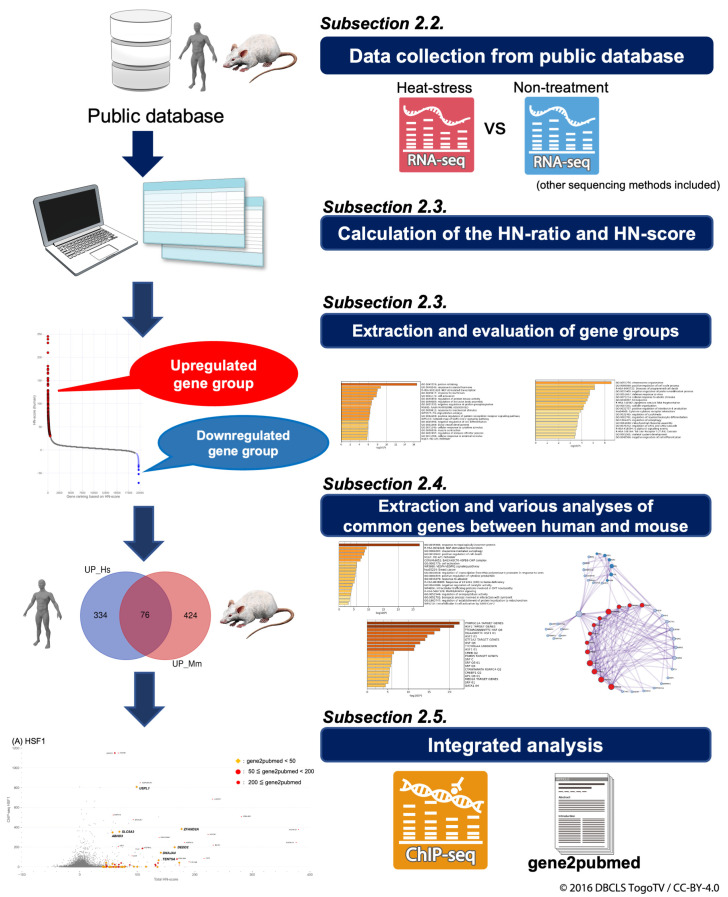
Flowchart of the analysis process. The flow of the analysis is shown, along with the analyses and figures performed in each section.

**Figure 2 ijms-24-13444-f002:**
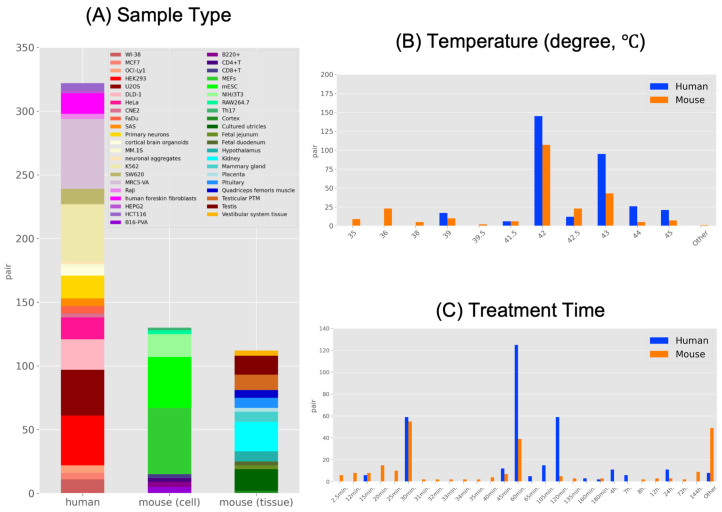
Summary of the contents of the heat stress-related dataset. (**A**) The stacked bar graph shows the number of data pairs collected from the sample. From left to right: human cell, mouse cell, and mouse tissue types. (**B**,**C**) The dataset summary for (**B**) temperature condition (degree, °C) and (**C**) treatment time.

**Figure 3 ijms-24-13444-f003:**
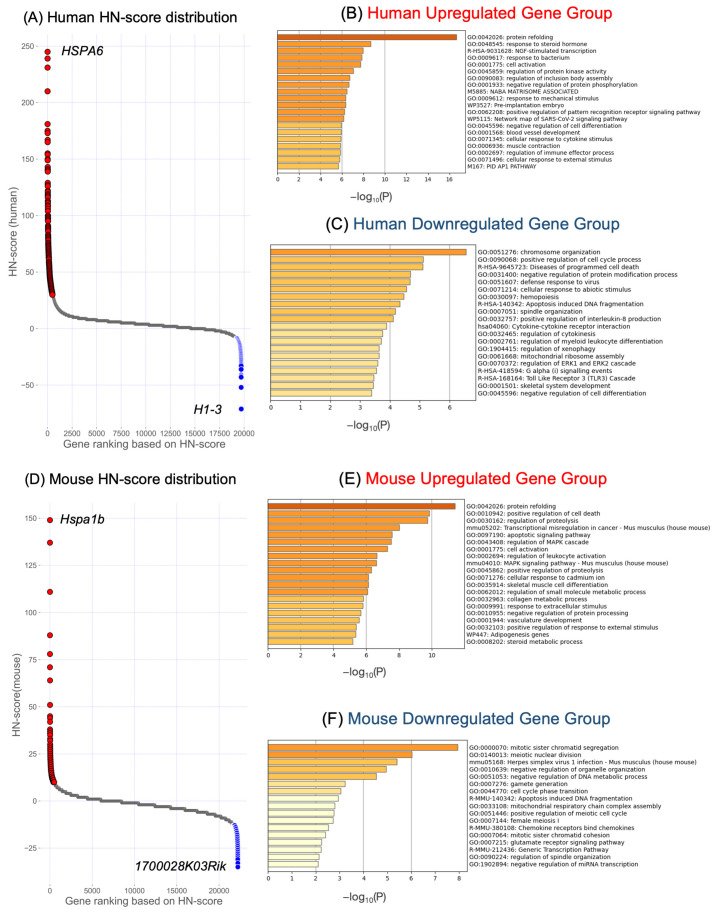
Scatter plots of the HN-score and gene set enrichment analysis of upregulated and downregulated human and mouse genes. (**A**,**D**) Scatter plots of HN-scores for all genes in (**A**) humans and (**D**) mice. Red and blue dots represent upregulated gene group (500 genes with high HN-scores) and downregulated gene group (500 genes with low HN-scores). Genes shown at the top right and top left indicate genes with the highest or lowest HN-score. (**B**,**E**) Results of gene set enrichment analysis of upregulated genes in (**B**) humans and (**E**) mice. (**C**,**F**) Gene set enrichment analysis of the downregulated genes in (**C**) humans and (**F**) mice.

**Figure 4 ijms-24-13444-f004:**
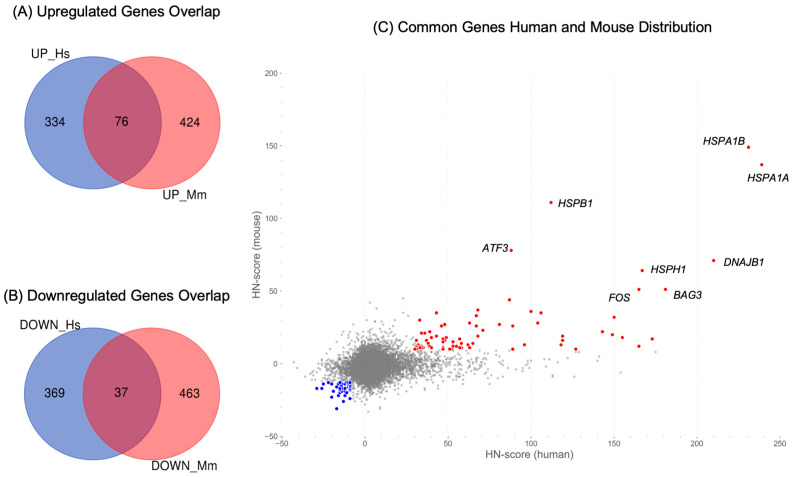
Overlaps of 76 upregulated and 37 downregulated human and mouse genes using mouse gene symbols. (**A**) Venn diagram of the upregulated genes in humans (UP_Hs) and mice (UP_Mm). (**B**) Venn diagram of downregulated genes in humans (DOWN_Hs) and mice (DOWN_Mm). (**C**) Scatter plots of common upregulated and downregulated human and mouse genes. The *x*-axis represents the human HN-score, and the *y*-axis represents the mouse HN-score. The red plot indicates the 76 common upregulated genes, whereas the blue plot indicates the 37 common downregulated genes, with the upper right indicating a higher HN-score and the lower left indicating a lower HN-score. The gene symbols shown in the figure indicate genes with HN-scores of 50 or higher in humans and mice, respectively.

**Figure 5 ijms-24-13444-f005:**
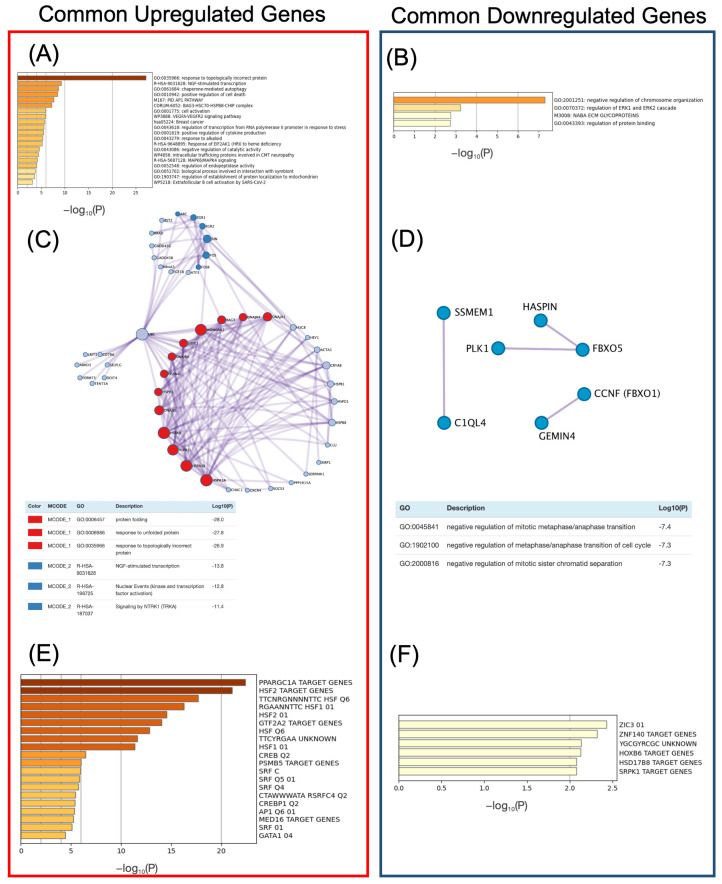
Various functional analyses of common upregulated/downregulated genes. (**A**,**B**) Gene set enrichment analysis of common (**A**) upregulated genes and (**B**) downregulated genes. (**C**,**D**) Protein–protein interaction network analysis of common (**C**) upregulated genes and (**D**) downregulated genes. (**E**,**F**) Transcription factor target gene enrichment analysis of common (**E**) upregulated genes and (**F**) downregulated genes.

**Figure 6 ijms-24-13444-f006:**
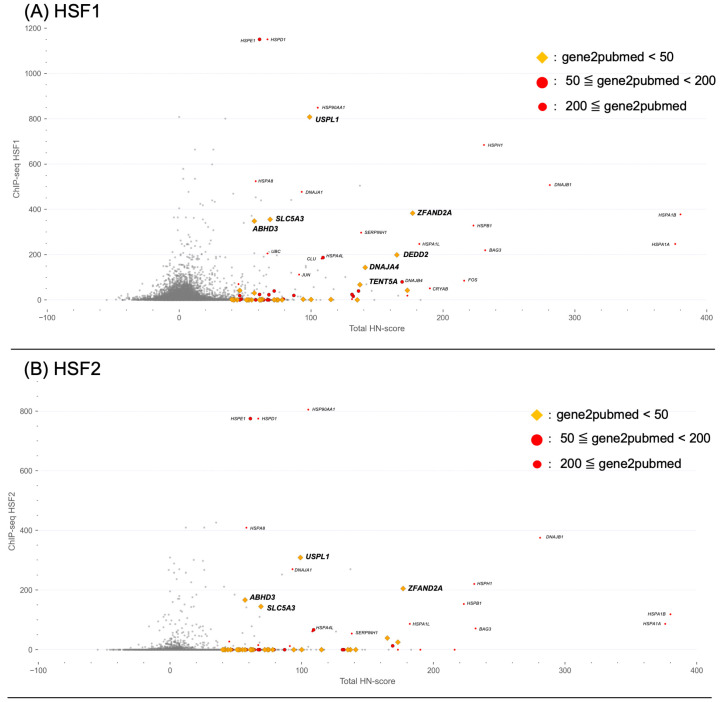
Integrated scatter plots using HN-score, average MACS2 score, and gene2pubmed data for several transcription factors. (**A**–**C**) Scatter plots from integrating the HN-score, ChIP-seq data, and gene2pubmed data. *X*-axis “Total HN-score” means (human HN-score) + (mouse HN-score). *Y*-axis indicates the average MACS2 score (peak mean) for the transcription factors (**A**) HSF1, (**B**) HSF2, and (**C**) PPARGC1A. Average MACS2 scores were calculated from human information. ChIP-seq data counts are (**A**) HSF1:105, (**B**) HSF2:25, and (**C**) PPARGC1A:1. Using gene2pubmed data, yellow diamonds were plotted against common upregulated genes if the number of reported papers was less than 50, large red circles if the number of reported papers was between 50 and 200, and small red circles if the number of reported papers was 200 or more.

## Data Availability

The data presented in this study are publicly available in figshare (https://doi.org/10.6084/m9.figshare.c.6564487) [21]. All the scripts used in this study are publicly available on GitHub (https://github.com/yonezawa-sora/HS_code, accessed on 1 August 2023).

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
