# Peer review of "Meta-Analysis of Heat-Stressed Transcriptomes Using the Public Gene Expression Database from Human and Mouse Samples"

_ijms, 2023, doi:10.3390/ijms241713444_

Round 1

Reviewer 1 Report

Comments and Suggestions for Authors

The authors tried to characterize heat stress-responsive genes using a meta-analysis of numerous gene expression data from the public database. And 76 upregulated and 37 downregulated genes were identified. Their findings are important for the further functional analysis of these upregulated genes. I provided some comments for the authors to consider as outlined below.

General comments:

1) Figures 1, 2, 4, and 5: The images are very vague and it is hard to see these fonts.

2) Line 114-118: “The upregulated gene group and downregulated gene group” were selected for the enrichment analysis and showed in the Figure 2. How about the results of enrichment KEGG pathway analysis of these genes?

3) Line 225-227: How about the results of transcription factor target gene enrichment analysis of these common downregulated genes?

Reviewer 2 Report

Comments and Suggestions for Authors

This study complements previously published efforts to document the effects of heat stress on cells. Great effort has been made to analyse a wide variation of datasets, and it is very useful to the research community that this data has been made available publicly. The content matches the impact factor of the journal. A couple of points should be addressed in my opinion to make this manuscript more attractive for publication:

1. The readability of the figures could be enhanced, and figures could be made more intuitive by either providing schemes of analysis or figure titles.

2. HN ratio’s and HN scores should be explained earlier in the text to improve readability.

3. The authors should refer to other studies that performed similar meta-analyses other than their own to broaden the scope of the paper and the context for the reader.

Comments on the Quality of English Language

-

Reviewer 3 Report

Comments and Suggestions for Authors

The authors have conducted a comprehensive study focused on gene expression analysis utilizing a diverse dataset. While the study presents a detailed account of gene expression data acquisition and analysis, several flaws and limitations should be acknowledged.

The text implies that genes common to both human and mouse in the upregulated and downregulated categories are directly indicative of shared heat shock response mechanisms. However, the mere presence of these genes in both species doesn't automatically imply that the same mechanisms are at play. Genetic, physiological, and evolutionary differences between human and mouse could lead to different regulatory responses even for the same genes.

The study assumes that because certain genes are shared between human and mouse in both upregulated and downregulated categories, these genes are part of common heat shock response mechanisms. However, these genes might be responding to different aspects of the stress in the two species or could be regulated by distinct pathways.

The primary flaw is the incorporation of gene expression data acquired using diverse sequencing methods, such as RNA-seq and Ribo-seq. This diversity introduces significant technical variability that can confound the study's findings. Different sequencing methods have distinct biases, error rates, and depth of coverage, potentially leading to inaccurate comparisons and interpretations.

The study's attempt to collect gene expression data from various sources with dissimilar sequencing methodologies compromises the standardization and comparability of the dataset. This might lead to difficulties in drawing robust conclusions from the data due to inherent variations introduced by these different methods.

The study does not adequately address the quality control and validation procedures performed on the collected gene expression data. Different sequencing methods can yield varying levels of data quality, and without a thorough validation process, the reliability of the data remains uncertain.

The focus on comparing heat shock (HS) conditions to non-treatment conditions overlooks the broader biological context. Gene expression changes can be influenced by numerous factors beyond heat shock, such as cell type, developmental stage, and disease status. The absence of such contextual information restricts the study's ability to draw meaningful insights.

The text in discussion doesn't sufficiently address the potential complexity of the heat shock response. Factors like the duration and severity of heat shock, cell type, and physiological state can all influence gene expression patterns. Failing to account for these factors might lead to an incomplete understanding of the underlying mechanisms.

While the study provides information about sample types, temperature conditions, and treatment times, it lacks details about other important factors that could influence gene expression, such as the physiological state of the cells, cell cycle stage, and specific stressors within the heat shock treatment.

The study integrates ChIP-seq data from an external database without detailing the methods used to ensure compatibility between the datasets. Combining data from different sources can introduce biases and errors if not properly accounted for.

The study appears to lack comprehensive statistical analyses to validate the significance of the observed gene expression changes and their associations with transcription factors. Without robust statistical tests, the study's findings might lack statistical rigor.

While the study identifies enriched transcription factor target genes, it doesn't establish a causal relationship between the identified transcription factors and gene expression changes. Correlation does not necessarily imply causation, and additional experimental validation is required to confirm the regulatory roles of these factors.

The study mentions that source data can be obtained from a specific repository, but it's important for transparency and reproducibility that all data and analysis code are made available in a standardized and accessible manner.

The study seems to rely heavily on computational analyses without thorough biological interpretation. Understanding the functional implications of the identified gene expression changes and their relation to heat shock requires deeper exploration.

While the text mentions protein-protein interaction network analysis and gene set enrichment analysis, it's unclear whether these findings were experimentally validated to confirm their functional relevance. The presence of genes in certain pathways or networks doesn't necessarily confirm their involvement in those functions.

Comments on the Quality of English Language

 Moderate editing of English language required
